# Urban Vulnerability Analysis Based on Micro-Geographic Unit with Multi-Source Data—Case Study in Urumqi, Xinjiang, China

Jianghua Zheng [1,2,†], Danlin Yu [3,*,†], Chuqiao Han [1] and Zhe Wang [1]

1 College of Geographical and Remote Sensing Science, Xinjiang University, Urumqi 830046, China; zheng.jianghua@xju.edu.cn (J.Z.)
2 Key Lab of Smart City and Environmental Modelling, Xinjiang University, Urumqi 830046, China
3 Department of Earth and Environmental Studies, Montclair State University, Montclair, NJ 07043, USA
* Correspondence: yud@montclair.edu
† These authors contributed equally to this work.

**Abstract:** This study introduces a novel approach to urban public safety analysis inspired by a streetscape analysis commonly applied in urban criminology, leveraging the concept of micro-geographical units to account for urban spatial heterogeneity. Recognizing the intrinsic uniformity within these smaller, distinct environments of a city, the methodology represents a shift from large-scale regional studies to a more localized and precise exploration of urban vulnerability. The research objectives focus on three key aspects: first, establishing a framework for identifying and dividing cities into micro-geographical units; second, determining the type and nature of data that effectively illustrate the potential vulnerability of these units; and third, developing a robust and reliable evaluation index system for urban vulnerability. We apply this innovative method to Urumqi's Tianshan District in Xinjiang, China, resulting in the formation of 30 distinct micro-geographical units. Using WorldView-2 remote sensing imagery and the object-oriented classification method, we extract and evaluate features related to vehicles, roads, buildings, and vegetation for each unit. This information feeds into the construction of a comprehensive index, used to assess public security vulnerability at a granular level. The findings from our study reveal a wide spectrum of vulnerability levels across the 30 units. Notably, units X1 (Er Dao Bridge) and X7 (Gold Coin Mountain International Plaza) showed high vulnerability due to factors such as a lack of green spaces, poor urban planning, dense building development, and traffic issues. Our research validates the utility and effectiveness of the micro-geographical unit concept in assessing urban vulnerability, thereby introducing a new paradigm in urban safety studies. This micro-geographical approach, combined with a multi-source data strategy involving high-resolution remote sensing and field survey data, offers a robust and comprehensive tool for urban vulnerability assessment. Moreover, the urban vulnerability evaluation index developed through this study provides a promising model for future urban safety research across different cities.

**Keywords:** micro-geographic unit; urban vulnerability; vulnerability assessment; multi-source data; urban public safety; Urumqi Tianshan District; urban public safety





## 1. Introduction

Urban spaces are highly concentrated locales for people, wealth, information, and built infrastructures, not to mention that urban spaces are also the most important space in the lives of humankind. Over half of the world's population is now living in cities. China is catching up fast.

As a matter of fact, urbanization in China was over 50% by the year 2015 [1]. While the rapid urbanization is certainly a sign of rapid socioeconomic development witnessed in China during the past four decades, cities in China are also experiencing increasing pressure

on the fronts of eco-environmental degradation [2–6], infrastructural and public service shortage [7–9], and public and individual safety and security concerns [1,10,11]. Among all the pressure, understanding urban vulnerability towards a variety of common threats, such as urban crimes, urban fire disaster, urban traffic incidents, and urban environmental pollution, has become increasingly important to not just the city governments and planners but to the urban dwellers, business owners, insurance underwriters, and potential investors as well.

Studies on urban vulnerability at the city scale have seen growing interests in the past decades [10,12–15]. Scholars attempt to establish a series of relationships between the possibility of the occurrence of various disasters and a variety of the cities' social, economic, physical, and environmental characteristics via statistical approaches [10,13,15,16]. Apparently, such studies would provide a quite useful understanding at a national/regional scale for policy makers to have a general view that will indicate which cities are prone to certain disasters, as adequately shown in [15].

Those studies, however, offer little help for city dwellers, a specific city's government, planners, business owners, insurance underwriters, and other interested parties to understand how vulnerable their daily lives would be based on their surroundings. Such understanding is an essential part of a vibrant, sustainable, and prosperous urban landscape since it constructs the ideal cornerstone of a livable and safe place for urban dwellers [10,17–19]. This current study hence for the first time embarks on such a task that investigates the characteristics of specific surroundings and how such characteristics can be generalized to quantitatively identify the specific surroundings' potential vulnerabilities via a simple generalization. In this research, we propose a term that characterizes the specific surroundings as the "micro-geographic unit" of a city since they refer to a relatively uniform, homogeneous build-up of neighborhood areas that are at sub-city/street level. This concept is one of the foundations of our current research. It represents a fresh attempt to model urban public safety from a more granular perspective that might enable better urban management and public safety measures.

Although conceptually the idea of a micro-geographic unit of a city is very appealing, two problems immediately arise. First, how can a micro-geographic unit be identified/demarcated in a city? Second, what information/data can be collected and summarized to describe the potential vulnerability of a micro-geographic unit?

Since the micro-geographic unit is a dynamic geographic concept, regular data collection strategies that are often suggested in vulnerability studies [10,12–15] might fall short to provide sufficient data. New strategies based on feature extraction of remote sensing images are more practical and reasonable to provide analyzable data to generalize the vulnerability scenarios of micro-geographic units.

For the first time, the current study attempts to address these two concerns in urban vulnerability studies at the sub-city/street level with the help of remote sensing information extraction and a case study in Urumqi, Xinjiang, China. We hold the belief that as an important urban public safety landscape identifier, the micro-geographic unit shall be treated as a dynamic instead of a static unit in that its boundaries are changing relatively frequently in response to changed surroundings. The identification of micro-geographic units of a city hence requires constant adjustment and shall be the result of a combination of the "general understanding of urban vulnerability landscape" and "local knowledge." Fieldwork is of utmost importance in identifying the micro-geographic units in a city, and frequent updating will be required to adjust the micro-geographic units to the changed surroundings. However, initial identification can be carried out through identifying key locations in a city and demarcated through a Voronoi diagram. In addition, with the help of remote sensing information extraction, we can then extract analyzable landscape features for those micro-geographic units. Using these extracted features, we will then build an index set for urban vulnerability measurement purpose.

In Section 2, we first highlight the significance of micro-geographical units in the study of urban vulnerability while proposing a Voronoi demarcation approach rooted in

fieldwork and local knowledge for their identification. We then provide an overview of our study area, the Tianshan District of Urumqi City, Xinjiang, China and proceed to discuss the development of an urban vulnerability index based on feature extraction from remote sensing images. We detail how such information is processed and how the vulnerability index is constructed at the micro-geographical unit level. Section 4 presents the results, specifically the features extracted from remote sensing images and the distribution of vulnerability indices across the micro-geographical units, along with a comprehensive discussion. Section 5 concludes our study, summarizing key findings and outlining future research directions.

## 2. Materials and Method

### 2.1. Study Area

Urumqi, the capital city of Xinjiang Uyghur Autonomous Region, one of the largest megacities in Northwestern China, was chosen as our case study. Mainly due to logistical reasons and funding availability, personnel, and time constraints, our study area focuses specifically on the built-up area of Tianshan District, Urumqi (Figure 1). That way, our field survey team was able to cover the entire built-up area for the 10 days of the survey. The choice of study area is both practical and convenient. On the one hand, as one of the largest megacities in Northwestern China, Urumqi has experienced rapid development during the past decade. GDP per capita increased from 14,622 CNY in 2000 to 69,865 CNY in 2016. The population increased from 1.82 million in 2000 to 2.68 million in 2016. Total road length increased from 994 km in 2000 to 1314 km in 2016. The number of vehicles also increased from 0.13 million in 2000 to 0.94 million in 2016 (*China Statistical Yearbook*). Consequently, problems accompanying rapid urban development also emerged as the city grew. On the other hand, different from many other Chinese cities in the east and central parts, Urumqi is also a city of mixed ethnicities with both Han and Uyghur people as the major ethnical groups, along with many other ethnic minorities. The 5 July 2009 terrorist attack in Urumqi evoked a major public safety concern of how to maximally reduce property and human life losses in mixed-ethnicity cities that are prone to internal terrorist attacks. Media coverage and reports are abundant, but studies from the morphology of the cityscape and perspective of small neighborhood-like micro-geographic unit remain limited.

### 2.2. Micro-Geographic Unit

2.2.1. Conceptual Origin of Micro-Geographic Unit

The micro-geographic unit proposed in this study is similar to the concepts of a streetscape or neighborhood in the urban studies literature [20–22] or simply referred to as a "hot spot" or "crime places" in criminology studies [23–31]. The latter tends to focus on specific "street segments" [31]. From an urban public safety perspective, the importance of micro-geographic units is more akin to the definitions used in urban criminology studies. Indeed, criminologists are among the earliest scholars who are interested in the idea of place and its role in the production and prevention of crimes [28,31,32]. As early as in the early 19th century, European scholars examined the distribution of crime across large administrative areas and created official government record keeping for better city management and governance [31,33].

In recent studies, criminologists became more interested in how crimes vary across different communities and neighborhoods and what might contribute to such variations [31,34,35]. Specifically, there is a growing interest among criminologists to look at the distribution of crime in smaller geographic units of a place such as addresses or street segments, or clusters of these micro units of geography [26,27,36,37].

An important catalyst for this interest came from theoretical perspectives that emphasized the context of crime and the opportunities presented in smaller geographic units to potential offenders [31]. Such "opportunities" we term as "micro-geographic unit elements" in this study (it is noteworthy here that "micro-geographic unit elements" are not necessarily related to urban crime but can be extended to other urban threats, such

as traffic accidents, fire disasters, environmental pollution, terrorist attacks, and the like; the set of such elements might vary depending on which threat is under study and how data are obtained, but the basic arguments stand) tend to vary from place to place in very subtle ways.

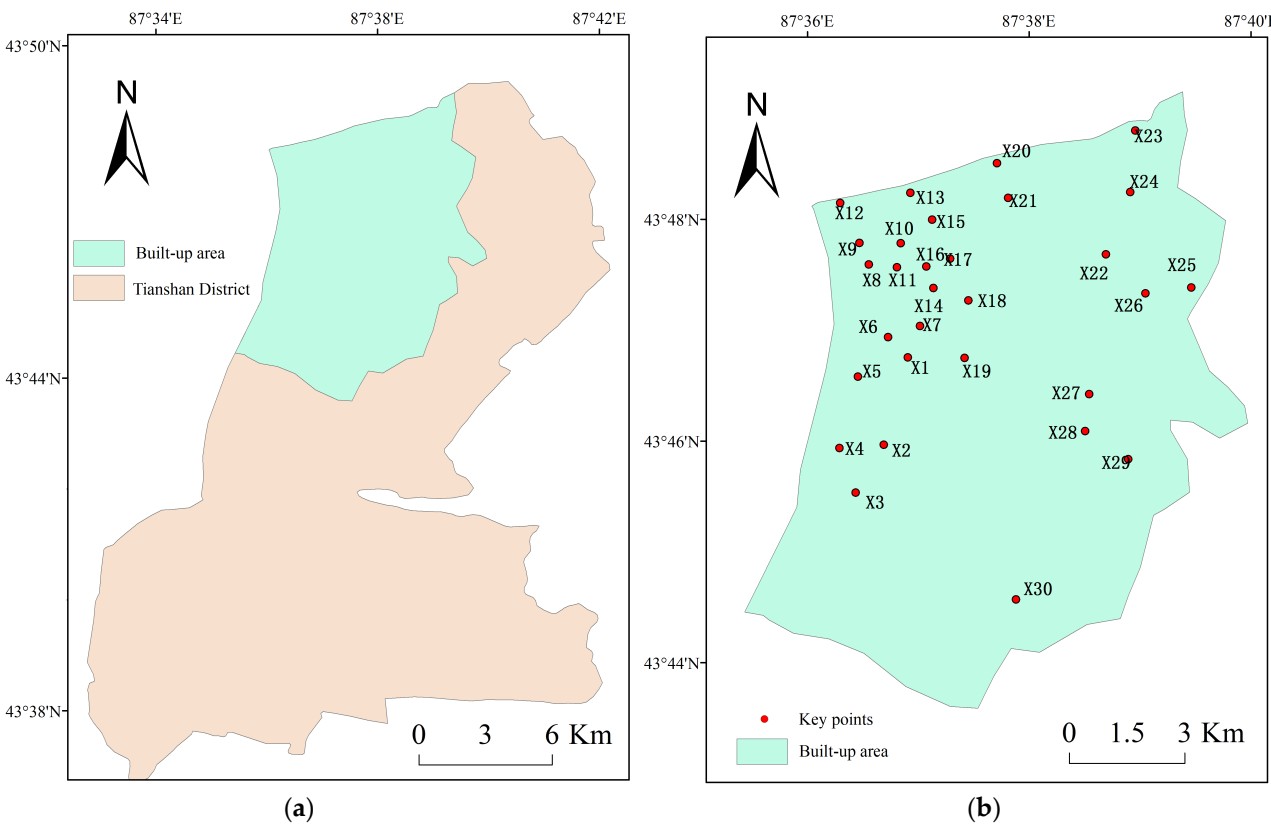

**Figure 1.** (**a**) The study area and the Tianshan District of Urumqi City, Xinjiang, China. (**b**) Survey locations on the map.

Studies at a coarser geographic scale could easily overlook such subtle variations and mask potential relationships that might exist between these "opportunities/elements" and incidents of various threats to urban public safety (such as traffic accidents, urban crimes, environmental pollution, potential terrorist attack, and other threats). One of the purposes of the current study is to define such a micro-geographic unit from the perspective of understanding and potentially quantifying urban vulnerability.

2.2.2. Field Work to Identify Micro-Geographic Units Based on Local Knowledge

A clear consensus definition of a micro-geographic unit is often missing because such a definition is usually dictated by the research purposes [10,37]. Under a larger urban public safety/health sphere, such small geographic units are usually specific locations within relatively larger social units of communities and neighborhoods [37]. In some studies, they are as small as a building or an address [38,39]; in other studies, they extend to block faces or street segments [27,31,36,40], still, others could include clusters of addresses, block faces, and streetscapes within a street block or even a few street blocks [26,41,42].

In this study, we adopt an exploratory approach to identify micro-geographic units for understanding, evaluating, and monitoring urban public safety, with urban crime and urban traffic accidents as two representative and common threats to public safety. Specifically, given our reliance on non-traditional data sources such as remote sensing image analysis and potentially web data mining (although not implemented in this study), our

strategy for demarcating micro-geographical units strictly adheres to a "local knowledge-based field survey" approach.

The routine followed a strictly defined design that centered on "local knowledge". From 13 to 30 July 2015, for every other day, a team that consisted of at least 4 students went out to various locations in Tianshan District, Urumqi for a field survey.

The team included two local students who have lived their entire life (up to the point of the fieldwork) in Tianshan District, Urumqi and other students who have studied in Xinjiang University (also located in the Tianshan District) for at least three years. The survey candidate locations were first selected on a map during a pre-survey conference held on 12 July. The input from the two local students was taken as high priority suggestions. All the team members contributed their comments based on their own observations and experiences of walking through the district.

The locations were selected based on four general principles. First, the selected locations must be relatively central to its surrounding areas so that residents and local activities in their immediate neighborhood traverse the locations often. Second, the selected locations must be relatively separated so that they have sufficient coverage of their immediate neighborhoods (we did not feel like it would be appropriate to focus on streetscape or small street segments due to the relatively complex mixed use of both residential and commercial landscapes in a typical Chinese city such as Urumqi). Third, the selected locations must have both convenient walking and automobile accessibility so that they are potentially maximally exposed to their immediate neighborhoods in a relatively homogeneous way. Fourth, the selected locations shall not be on the borders of the Tianshan District to avoid possible insufficient coverage of the border areas.

Based on the four grand principles and consultation with the local students, we had initially identified 133 survey locations on the map. After the actual field work started, we realized that the number of survey locations was too dense, and the suburban areas of the Tianshan District were inconvenient for both walking and automobile access. The adjusted survey strategy eventually reduced the number of survey locations to 30 and focused only on the built-up areas of the Tianshan District (Figure 1). These locations are as follows:

X1: Erdaoqiao International Bazaar;
X2: Xinjiang University;
X3: Water Park–Nanjiao Passenger Station;
X4: Urumqi South Park;
X5: Sinopec Gas Station/Public Transport Stop;
X6: Autonomous Region People's Hospital;
X7: Golden Coin Mountain International Plaza;
X8: Danlu Fashion Department Store;
X9: Chenggong Square;
X10: People's Cinema (roundabout five-way intersection);
X11: Baihua Village Computer City;
X12: Century Golden Flower Times Square;
X13: Xinjiang Education College;
X14: People's Theater Night Market Leisure Square;
X15: Urumqi First People's Hospital;
X16: Big Cross;
X17: People's Square;
X18: Nanmen International City;
X19: East Ring Integrated Market;
X20: Orthopedic Hospital;
X21: Corps Hospital;
X22: Xingfu Road Meteorological Community;
X23: Sun, Moon, and Stars Light Garden;
X24: Urumqi Fourth Hospital;
X25: Vocational University;

X26: Happiness Flower Court;
X27: Changle Garden;
X28: West Region International Trade City;
X29: Morning Light Garden;
X30: South Campus of Xinjiang University.

The fieldwork was repeated (in a shortened period) again in July 2016 to decide whether adjustment would be needed. The shortened fieldwork found our previous definition of the micro-geographic units still workable, and data could be steadily assembled for them. The research roadmap and framework are shown in Figure 2.

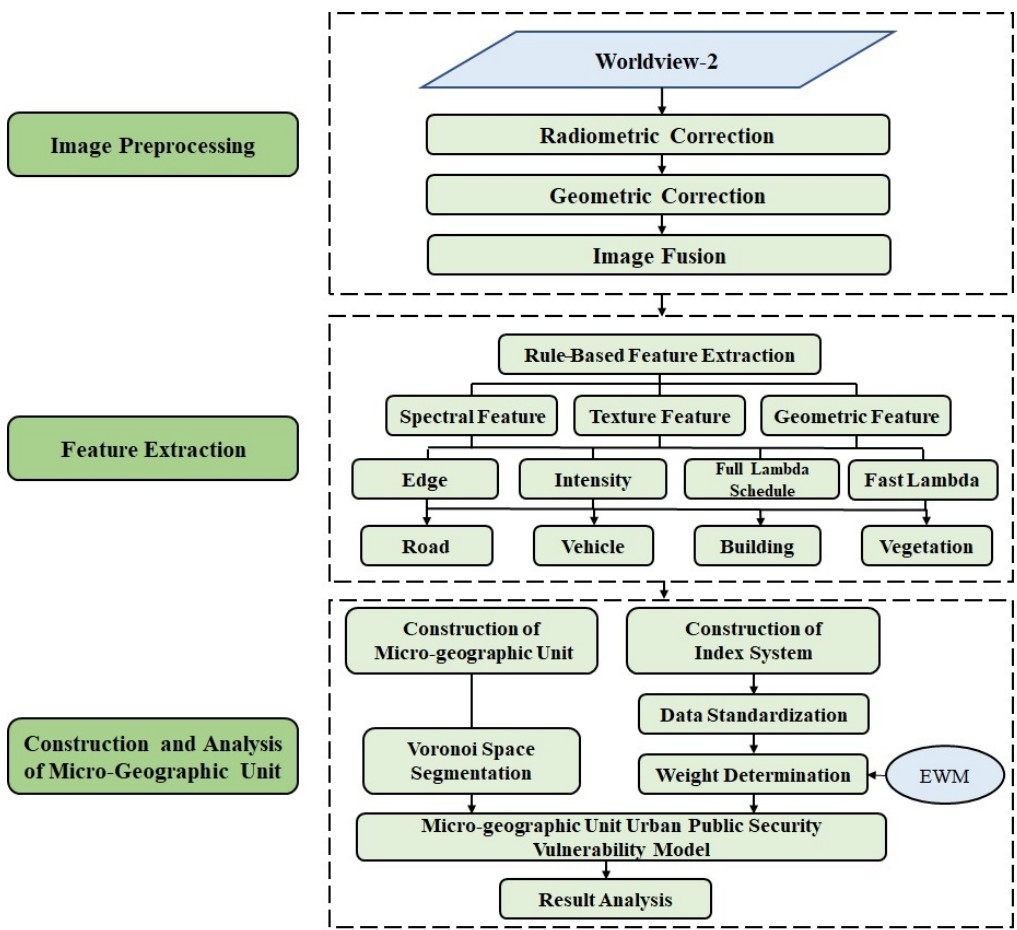

**Figure 2.** Technical framework of this study (EWM stands for entropy weighting method).

### 2.2.3. Defining Micro-Geographic Unit

One of the key research items in the current study is a clear definition of the micro-geographic unit. As aforementioned, while a consensus of how micro-geographic units can be defined is not yet available, we attempted to do so by adhering to a "local knowledge-based field survey" approach. Our fieldwork identified 30 key locations. Based on these 30 key locations, and recognizing the diversity of a city's internal spaces, we partitioned the city district into 30 distinct micro-geographic units.

Specifically, with the boundary of the study area (the Tianshan District's built-up area) and the 30 key locations, we constructed a Voronoi diagram (Figure 3a) to recognize from a spatial perspective that every location within the micro-geographic unit shall be closest to one of the key locations that defines the micro-geographic unit.

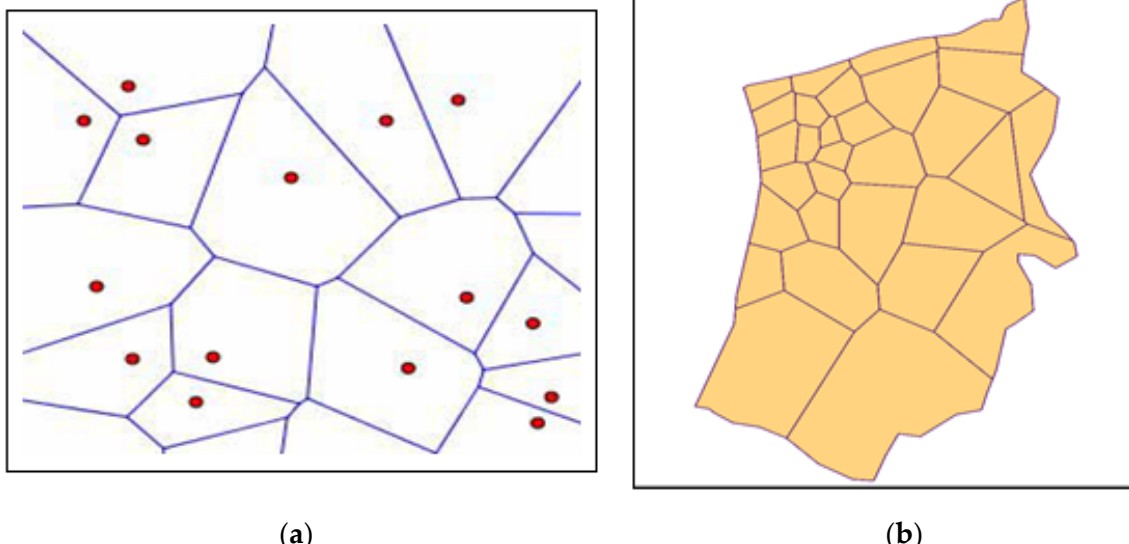

<div align="center">(<b>a</b>)          (<b>b</b>)</div>

**Figure 3.** (**a**) Voronoi graph (the red dots are locations where the Voronoi graph is generated—the distances between any points within each Voronoi block and these red dots are the shortest comparing to the other red dots). (**b**) Distribution of micro-geographic units in the built-up area of Tianshan District.

Once these Voronoi spatial units were initially identified, we procured high-resolution satellite remote sensing images for Urumqi. We overlaid these images onto the Voronoi spatial units to create the initial micro-geographic units. Then, we would consult the local surveyors' opinions as well as field investigation results and make appropriate adjustments to the original Voronoi spatial unit boundaries. Specifically, we adjusted the Voronoi boundaries using the nearest streets so that the identified micro-geographic units were consistent with street-separated communities. The final micro-geographic unit map ensureed maximum spatial anisotropy among one another, while maintaining basic homogeneity within the units accommodating to the existing roads and landmark boundaries (Figure 3b). Defining these micro-geographic units clearly forms the basis of our research and represents a focal point and innovative aspect of our work.

### 2.3. Development of Micro-Geographic Unit Based Urban Vulnerability Index

2.3.1. Urban Vulnerability Studies in the Literature

Vulnerability study has mainly focused on biophysical/social vulnerability and vulnerability of a built unit [12,13,43]. The vulnerability of a micro-geographic urban unit, however, largely eludes both the criminological and urban studies literature due to the complexity of urban space, especially at the micro-geographic level. Regarding certain urban threats, how vulnerable a specific location is shall be determined by the characteristics of such a location. For instance, a neighborhood with a high density of residents/buildings will be more vulnerable to contagious diseases than neighborhoods with a lower density. Communities having a high percentage of wood structure buildings are more prone to fire danger than the ones with brick structure buildings, yet the latter might suffer more from potential seismic damage than the former. To have a manageable understanding of urban vulnerability towards certain threats, the first step would be to generate a synthesized index that considers various micro-geographic units' elements (the "opportunities") that potentially contribute to threats endangering urban public safety.

Such elements are both abundant and different regarding different threats. Early studies identify them as some specific aspects of urban design [44] and urban architecture. With growing incidents of various threats, especially in urban crime, studies start to take into account a much larger set of characteristics embedded in the physical space and societal

unit that might provide potential opportunities for specific crime or other urban public safety threats [25–28,31,32,36,37,40,45–48].

Brantingham and Brantingham [48] classified the micro-geographical units into two broad categories with their data from Cambridge, England: crime generators and crime attractors. While the former includes hot spots that provide plenty of opportunities for crime, such as shopping and entertainment areas where crowds are common, the latter, on the other hand, includes spots in which offenders seek out victims in a planned and deliberate way, such as bar districts, prostitution areas (red lamp areas), and drug markets that are often deemed less guarded by authorities. Such categorization leads to an interesting observation that certain micro-geographic elements (such as easiness for crowdedness and concentration of alcohol, tobacco, or other selling facilities for addictive products) might be perceived by potential offenders as attractive. Micro-geographic units that are abundant with such elements would then be rather vulnerable towards certain crime. Studies attempting to determine relationships among occurrence of different crime types and distribution of various crime-attracting elements are often seen in the criminology literature [49–52]. Other urban public safety threats, such as an urban fire, traffic accident, environmental disaster, and home tenure insecurity and how the micro-geographic unit elements influence them can be extended in a similar way but are less studied. From an urban public safety perspective, although urban crime (especially violent crime) is a particularly severe threat, other threats need to be addressed as well. For this purpose, we attempted to develop a so-called urban public safety vulnerability index based on specific micro-geographic unit elements. The ideas are similar to urban criminologists' approaches of observing and obtaining useful information regarding the potential micro-geographic units and generating a synthesized composite index that could provide useful information for urban planning and urban development.

Developing an index for urban vulnerability requires a clear understanding of the concept of urban public safety to ensure relevant data collection. Recognizing that urban public safety encompasses a vast array of connotations, we have narrowed the scope for practicality in this study. We started our journey of understanding urban vulnerability based on the UN's Habitat Report 2007 [10,53,54], which outlines three main threats to urban public safety: urban crime, housing tenure insecurity, and natural or man-made disasters.

Our study adopts strategies from criminology studies to identify potential micro-geographical elements that serve as crime attractors or generators. Following Cutter, Mitchell, and Scott's [12] study of vulnerability, we appreciate that hazards include socially constructed situations, and thus, our vulnerability study should recognize a comprehensive set of biophysical, building, environmental, social, economic, and demographic indicators. In other words, our vulnerability study takes into account not only the threats causing place vulnerability but also the specific contexts in which these threats occur.

However, the elements influencing vulnerability can vary based on the specific threats to urban public safety. Beyond the three general categories of urban public safety threats as identified by the UN, the fundamental urban structural, demographic, and building-environment elements like street length, street complexity, population density, and building density are important in constructing vulnerability towards urban threats as they form the basis of an urban micro-geographical unit. The ultimate aim is to construct an urban vulnerability index that can be used to create an urban vulnerability-scape or risk-scape [12]. Such a risk-scape—mappable and providing a direct, visible, and manageable visual interpretation of urban vulnerability—could effectively guide responses to potential risks or threats.

In this study, following the core ideas of the UN's report, we focused on creating a vulnerability index considering two specific threats to urban public safety, namely urban crime and traffic incidents. Future research could add more threats to broaden the scope of responsible elements and create a more comprehensive index. The focus on urban crime and traffic accidents is instrumental, as these aspects constitute the most visible parts of urban public safety. A measure of vulnerability would provide immediate policy benefits,

enabling decision-makers to allocate resources effectively to combat crime and mitigate traffic accidents and to help city dwellers make informed decisions to avoid involvement in these incidents.

### 2.3.2. Data Items and Acquisition

A key component of any vulnerability assessment is the acquisition of systematic baseline data, particularly at the proposed micro-geographic unit level. This data provide inventories of hazardous areas and vulnerable population information that are essential for preimpact planning, damage assessments, and post disaster response. A vulnerability assessment requires not only an audit of potential hazards but also an understanding of the human dimensions involved. In the hazard study literature, three sets of information are usually required to assess vulnerability [12]: the identification of threats, threat frequency, and threat-zone delineation. As for the identification of threats, we have conducted intensive field surveys to identify potential threats. Our fieldwork identified that the conflicts between pedestrians and road traffic are one of the most frequent threats to urban public safety (Figure 4). For the majority of the time during our fieldwork, the conflicts were constantly present.

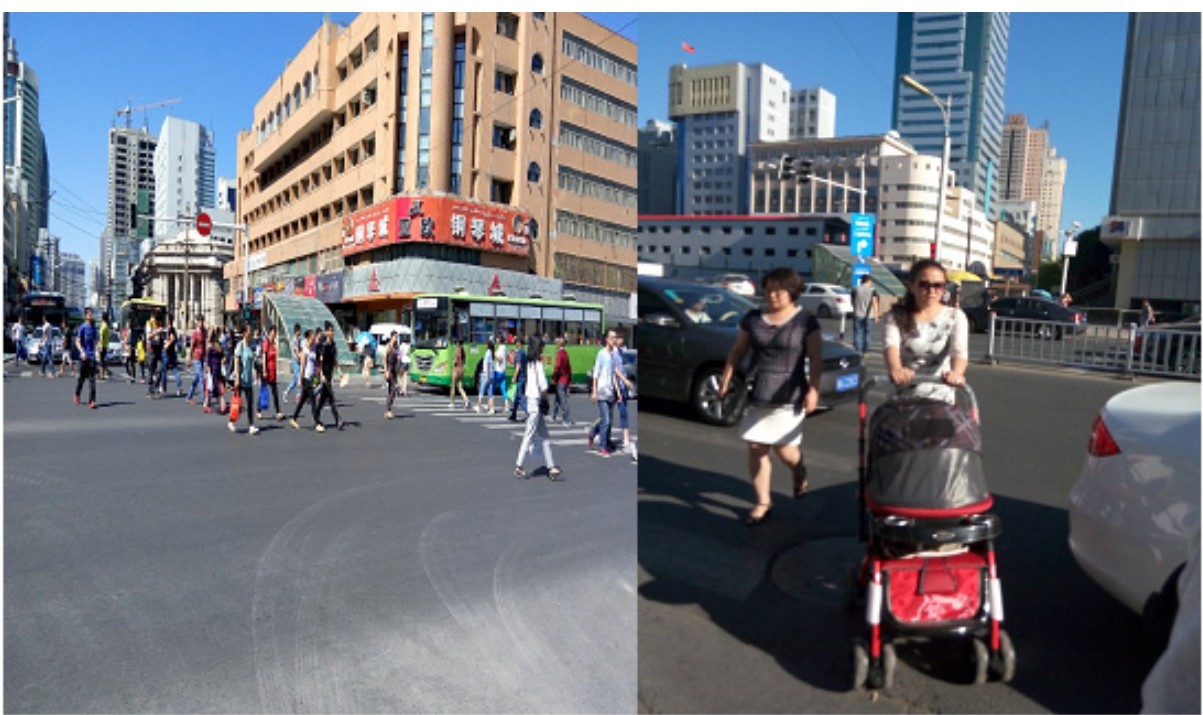

**Figure 4.** Frequent conflicts between pedestrians and road traffic.

The threat-zones are delineated at our designated micro-geographic unit level. Per the identified frequent threats to urban public safety, at the micro-geographic scale, we proposed an indicator scheme (or micro-geographic unit element set) that will contribute towards building the vulnerability index, and hence, quantify the threats for better management before, during, and after urban public safety incidents. More specifically, since we will be relying on acquiring information from remote sensing images and field surveys instead of traditionally available data sources, we proposed an indicator system of three categories with eight individual indicators.

The first category is the traffic and road category, which includes average road vehicle density (count of cars on roads/kilometer), vehicle density (count of cars/square kilometer), road density (kilometer/square kilometer), and road complexity (count of intersections/square kilometer). All four indicators were acquired from feature extraction analysis from remote sensing images. Among the four indicators, except for road

density, the other three indicators tended to be positively related to the micro-geographic units' vulnerabilities.

The second category is the building environment category. This category includes only one indicator, the building density (amount of buildings/square kilometer), which was also acquired with feature extraction from the remote sensing image. Building density was also positively related with the micro-geographic units' vulnerabilities. The argument stands that a densely clustered building environment often provides more potential hideouts for crimes and leads to less desirable road traffic conditions.

These two categories will essentially provide a relatively clear picture of the frequent pedestrian and road traffic conflicts, as we observed during our fieldwork.

The third category is the living quality category. This category includes three indicators, the vegetation patch density, vegetation extension index, and vegetation Shannon index. Except for the vegetation patch density, which suggests the degree of fragmentation of vegetation coverage, the other two indices could be linked to potentially reduce urban vulnerability by providing the residents a more comfortable, hence safer, living environment. Again, all three indices were acquired from feature extraction from the remote sensing image.

Remote sensing data, with its wide coverage, easy accessibility, and frequent updates, have become a crucial data source across various research fields. The WorldView-2 satellite, with an orbital height of 770 km and a revisit period of about 1.1 days, offers high-resolution data with a sub-satellite point resolution of 0.48 m, a spectral resolution of 1.8 m, and a swath width of 16.4 km. The geometric and texture features of its image data, owing to its high spectral, spatial, temporal, radiometric resolutions, and precise positioning accuracy, are particularly suited for extracting features relevant to our current study.

We chose WorldView-2 satellite image (Figures 5 and 6) coverage in the Tianshan District of Urumqi collected on 2 May 2015, with a cloud cover of 0, a minimum solar height of 60.28°, and the total area of 216 square kilometers. It includes 8 multi-spectral bands and a panchromatic band, each of which is relatively narrow, providing abundant information of ground objects.

During the imaging process, satellite images can be affected by various factors leading to geometric or radiometric distortions. For a quantitative analysis, it is crucial to preprocess these images. As such, we conducted radiometric correction, geometric correction, and image fusion to eliminate the geometric distortion, enhance the spatial resolution of the multi-spectral image, and retain the spectral information to meet our research objectives. By applying visual interpretation, we were able to determine the distribution of the built-up area in the Tianshan District.

### 2.4. Methods

### 2.4.1. Extracting Features Based on Object-Oriented Classification

The WorldView-2 remote sensing image was employed to extract information about vehicles, roads, buildings, and vegetation using the object-oriented classification method provided in the ENVI® software. This method was selected for its consideration of not only the spectral characteristics of the research object but also its size, shape, and texture. The equal spatial features of this approach proved highly effective for extracting features such as vehicles, roads, and buildings, which possess distinctive texture and shape attributes.

The object-oriented feature extraction process in ENVI software comprises two steps. The first step, "Find Object", involves selecting appropriate segmentation and merging scales for extracting different feature information. It is important to avoid features that are either too small or too large as much as possible. The second step, "Extract Features", involves analyzing the features of the categories by identifying their spectral features, texture characteristics, and other aspects to establish a suitable rule set. This process requires constant adjustments to ensure accuracy and precision.

### 2.4.2. Establish an Indicator Library

Building upon the object-oriented classification, we constructed an index library using the extracted feature information for vehicles, roads, buildings, and vegetation (See Table 1). This library incorporates the three categories and eight indicators discussed in Section 2.3.1. The calculation and interpretation of these specific indicators are detailed as follows:

**Table 1.** The specific meanings of indicator library.

| Micro-Geographic Unit Vulnerability Index with Attribute | The Specific Meanings of Indicators |
| --- | --- |
| Vegetation patch density (negative) | Fragmentation degree of patch |
| Vegetation extension index (positive) | Aggregation degree of patch |
| Vegetation Shannon index (positive) | The size and uniformity of patch area |
| Average road vehicle density (negative) | Vehicles per unit length of road |
| Vehicle density (negative) | Vehicles on roads per unit area |
| Road density (positive) | Road length per unit area |
| Road complexity (negative) | Reflected by the average road node degree, it refers to the count of intersections per unit area |
| Building density (negative) | Building coverage |

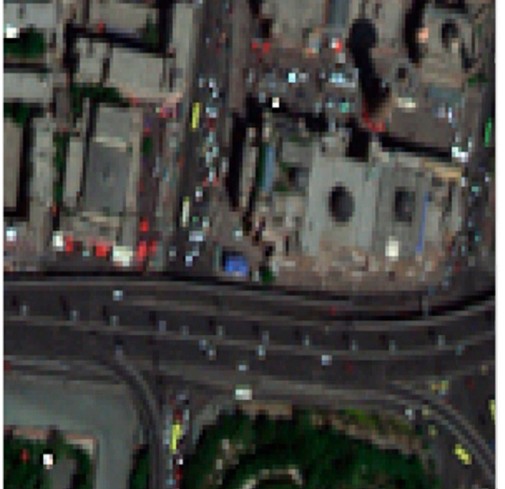
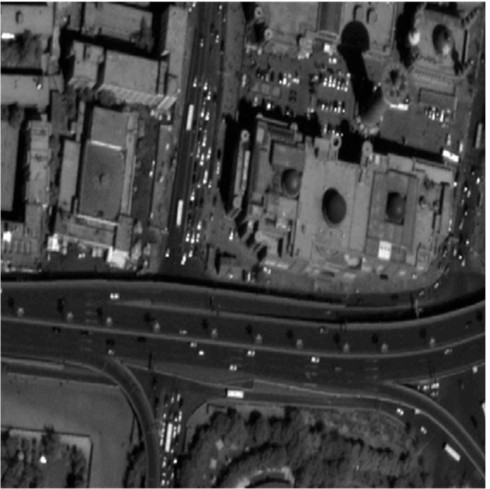

(**a**)  (**b**)

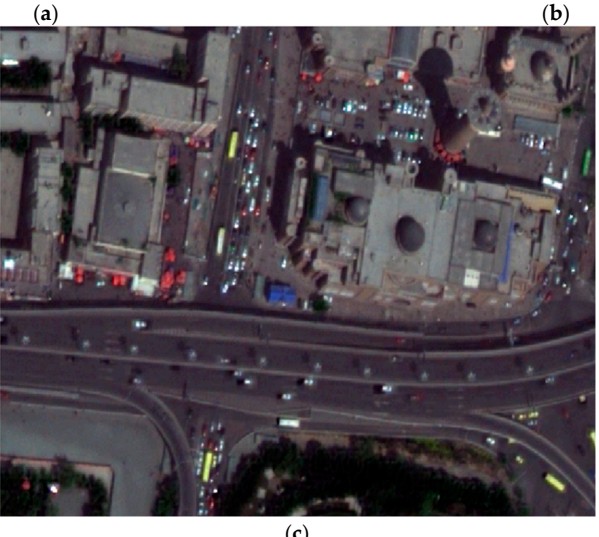

(**c**)

**Figure 5.** WorldView-2 (**a**) multi-spectral imagery (5,3,2 bands). (**b**) Panchromatic image. (**c**) Fused image (5,3,2 bands) in the study area.

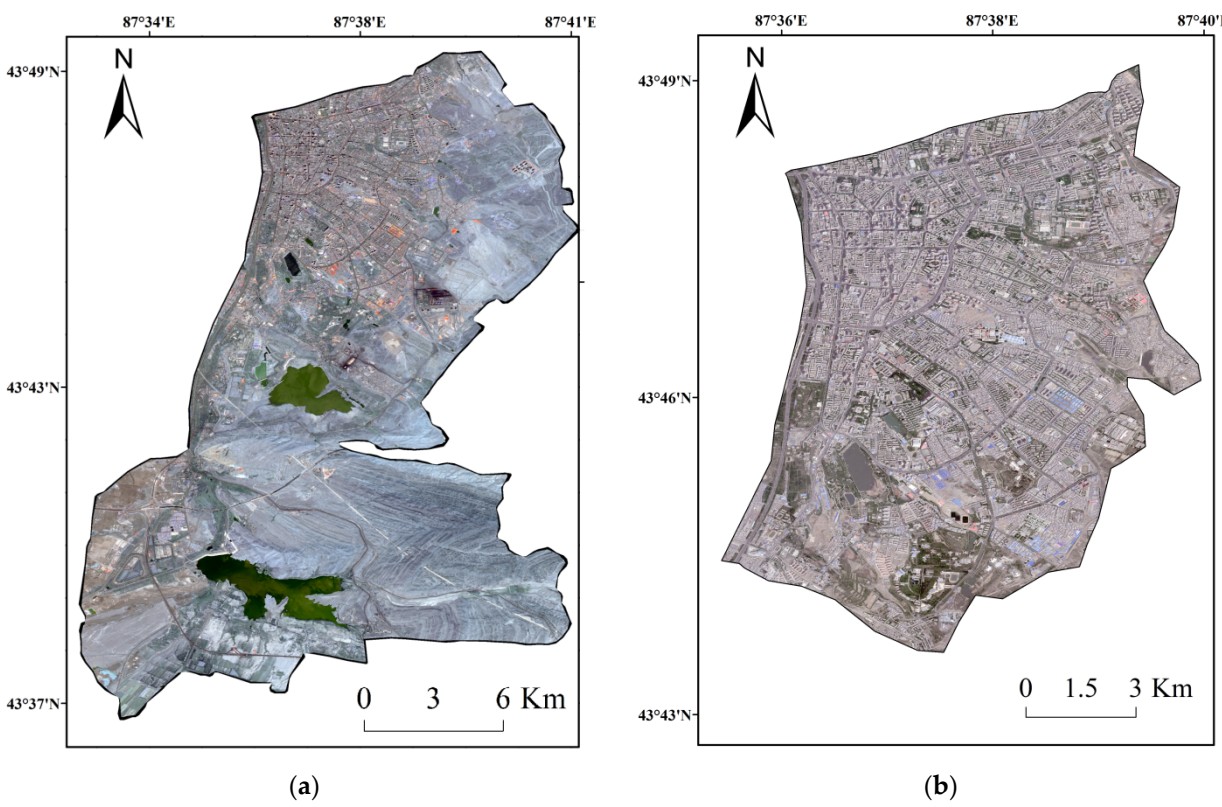

**Figure 6.** WorldView-2 (**a**) in Tianshan District image and (**b**) built-up area image.

### 2.4.3. Construction of Vulnerability Model
#### Data Standardization

To measure the vulnerability of the city, we must first standardize different data to remove the impact of the dimension. In the process of the vulnerability assessment, since the positive and negative indicators have different effects on the evaluation results, they need to be treated differently and calculated u different formulas. If there are *n* evaluation objects and *m* evaluation indicators, we use *i* for the evaluation object, *j* for the evaluation index, and $X_{ij}$ represents the *j*th index value of the nth evaluation object.

The positive Indicators calculation formula is

$$S_{ij} = \frac{X_{ij} - Min(X_j)}{Max(X_j) - Min(X_j)} \tag{1}$$

The negative indicators calculation formula is

$$S_{ij} = \frac{Max(X_j) - X_{ij}}{Max(X_j) - Min(X_j)} \tag{2}$$

In Formulae (1) and (2), $S_{ij}$ represents the normalized value of $X_{ij}$ after dimensionless processing, $S_{ij} \in [0, 1]$.

#### Determining Indicator Weights with Entropy Weighting Method (EWM)

In practice, indicators do not always contribute similarly to the final index. Weighting is often required to address this concern. There are two primary methods for determining index weights: subjective and objective weighting. Compared to the objective weighting method, the subjective method can be perceived as arbitrary and lacking in scientific rigor. Among the various objective weighting methods, the entropy weighting method (EWM) has gained widespread use in social and economic research fields. This is due to its

ability to reflect the effective value of index information and overcome the overlap between indicators [55,56]. More importantly, preliminary analysis of all the $S_{ij}$ suggest that there are correlations between the vegetation indices and the vehicle density indices. With the entropy weighing method, we will be able to generate weights that can effectively overcome the potential influence by the correlation among the non-weighted indices. Therefore, in this study, we employ the entropy weighting method to derive the weight coefficient of the urban public safety vulnerability evaluation index. The following are the calculation steps [56]:

(1) Based on the normalized data, $S_{ij}$, the proportion of $S_{ij}$ in $S_j$ of the $i$th sample under the $j$th index, is calculated, thereby constructing a matrix $P = [P_{ij}] n \times m$; The calculation formula is

$$P_{ij} = \frac{S_{ij}}{S_i} \tag{3}$$

(2) Calculating the information entropy of the index value $e_j$, the calculation formula is

$$e_j = -\frac{1}{\ln(n)} * \sum P_{ij} * lnP_{ij} \tag{4}$$

(3) The information entropy redundancy can be calculated using

$$g_j = 1 - e_j \tag{5}$$

(4) The weight of the indicator can be determined using

$$w_j = g_j / \sum g_j \tag{6}$$

In the above formulas, i = 1, 2, . . . , n; j = 1, 2, . . . , m.

Evaluation and Analysis Model

Our vulnerability evaluation model was developed based on urban vulnerability factors and their weights as detailed above. After calculating the index weights, we utilized this information to construct an urban vulnerability assessment model as follows:

$$UVI_i = \sum_{j=1}^{7} w_j * S_{ij} (i = 1, 2, \ldots, 31; \ j = 1, 2, \ldots, 7) \tag{7}$$

In formula (7), $w_j$ represents the weight of the $j$-th evaluation index; $S_{ij}$ represents the normalized value of the $j$-th indicator of $i$ micro-geographic units; and $UVI_i$ represents the degree of vulnerability of the $i$th micro-geographic unit.

### 2.4.4. The Classification of Vulnerability Level

At present, studies centered on urban vulnerability remain somewhat scarce, and there is no universally accepted standard for vulnerability classification. We have drawn upon relevant urban vulnerability research [10,15,18,57–64] discussed among the research team and combined the inputs from local students' life experiences. At the end, we have categorized the urban vulnerability index (*UVI*) into five levels (Table 2): high level ($0.8 \leq UVI < 1$), somewhat-high level ($0.6 \leq UVI < 0.8$), medium level ($0.4 < UVI \leq 0.6$), somewhat-low level ($0.2 \leq UVI < 0.4$), and low level ($0.0 \leq UVI < 0.2$).

**Table 2.** Vulnerability levels.

| Vulnerability | Low Vulnerability | Somewhat-Low Vulnerability | Medium Vulnerability |
|---|---|---|---|
| Vulnerability Index | 0—0.2 | 0.2–0.4 | 0.4–0.6 |
| States | Very good | Good | General |
| **Vulnerability** | **Somewhat-High Vulnerability** | **High Vulnerability** | |
| Vulnerability Index | 0.6–0.8 | 0.8–1 | |
| States | Alert | Crisis | |

## 3. Results

### 3.1. The Results of Feature Extraction

The feature extraction function of the ENVI software was employed to extract information pertaining to urban aspects such as vehicle targets, roads, buildings, and vegetation. We noted that the albedo of vehicles varied with changes in solar radiation and lighting conditions. Generally, vehicles can be categorized into two types: those with high brightness values (bright cars) and those with low brightness values (dark cars). The sun's altitude angle significantly influences vehicle shadows; however, in this image, the large solar angle of 70 degrees results in smaller shadow areas, thus minimally impacting our research.

We discovered that various types of vehicles could be segmented, and the darker sections of vehicles could be delineated under edge 85 and fast lambda 25 thresholds. Vehicles typically display a relatively regular distribution along roads or in parking lots and have distinct texture features that distinguish them from surrounding objects. However, the size and shape of vehicle silhouettes differ, contingent on the angle and intensity of solar radiation. Therefore, rule sets were established according to the various categories of vehicles. For instance, a bright car is defined as having a Spectral Mean (Band 1) greater than 130. The area of a vehicle, which is less influenced by sunlight than the actual size of the vehicle, is defined as falling within the range [2, 14].

The shapes of bright cars are typically irregular, influenced by the sun's angle, so their ductility is not considered in our assessment. Conversely, dark cars, with a brightness value ranging between 60 and 120, exhibit a more consistent rectangular shape on the map, unaffected by the sun's altitude angle. Their elongation is generally less than 2.34. Therefore, their area value range was set as 2.5 to 18. Figure 7 demonstrates the combined results for cars, with the final extraction result shown in Figure 8a.

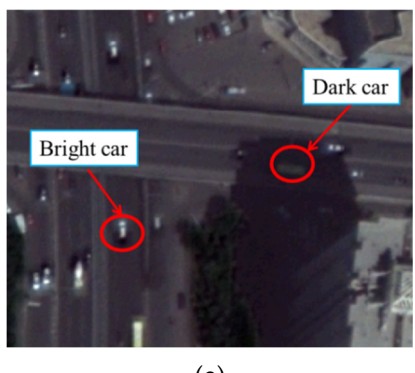 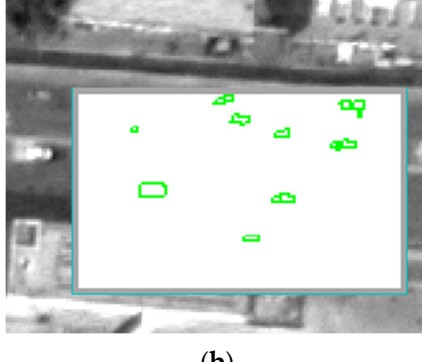 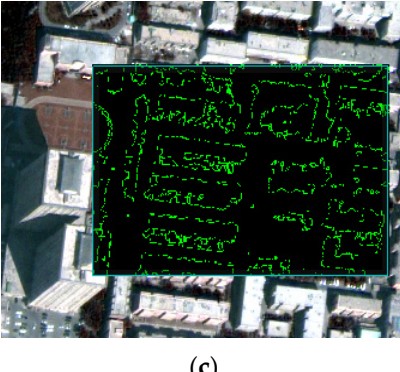

(**a**) (**b**) (**c**)

**Figure 7.** (**a**) The bright car and dark car. (**b**) Merged cars from the WorldView-2 image. (**c**) Merged and split buildings from the WorldView-2 image.

Roads, which typically form a cross-connected network, have spectral characteristics significantly different from those of surrounding objects. The image was segmented using ENVI software with a segmentation and merging scale of (40, 80). Afterward, road feature extraction was conducted based on image segmentation. Roads appear as long rectangles in the image, leading to substantial ductility. The elongation weight was assigned a maximum

value of 0.35, ranging from 23.00000 to 108.81313. The area is also extensive, ranging from 135.00000 to 480.00000 with a weight of 0.25. The length varies from 100 to 2000.0000, as illustrated in Figure 8b.

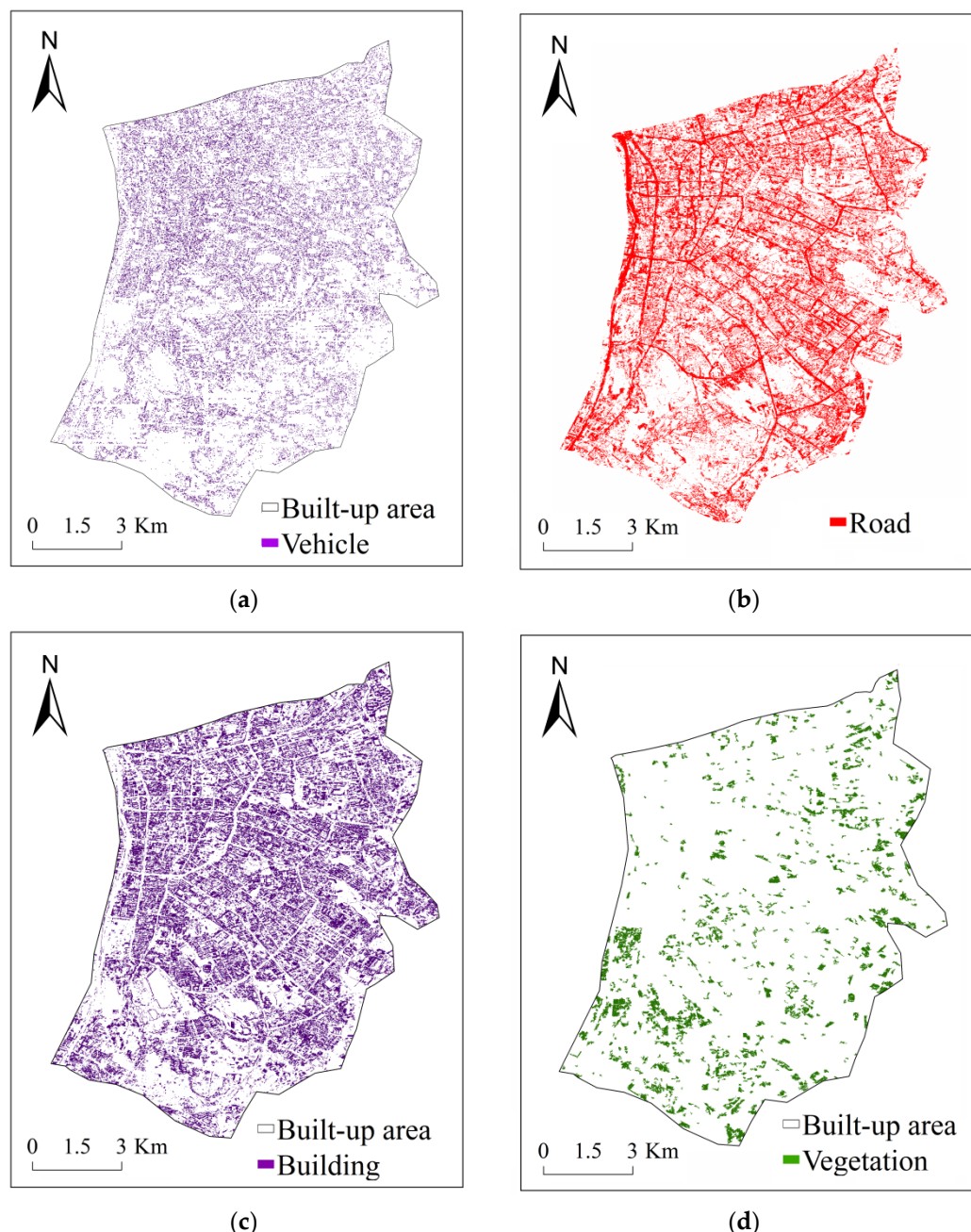

**Figure 8.** The feature extraction results of (**a**) vehicles (**b**) roads (**c**) buildings, and (**d**) vegetation from the WorldView-2 image.

The term "buildings" in this study encompasses various city high-rises and facilities such as factories. Buildings usually exhibit a set of characteristics, with most appearing as regular rectangles, having smaller areas and less ductility than roads. After multiple adjustments, Intensity40 and Fast Lambda90 were chosen as the combined segmentation scale, resulting in effective building segmentation. However, the extraction of building information can be influenced by vegetation, roads, and concrete, necessitating certain measures. We removed vegetation disturbance at Tolerance 5, eliminated road features when the area was greater than 50 and elongation less than 3 and removed concrete floor

interference when the Spectral Mean (GREEN) < 650. The result was a clear depiction of building information (Figures 7c and 8c).

To extract vegetation coverage, we used a combination of bands 4, 3, and 2 from the WorldView-2 images, making the vegetation appear red on the image. Vegetation extraction utilized the NDVI (Normalized Vegetation Index) method; a higher NDVI value implies more abundant vegetation. In this practice, a 35-segmentation scale and a combined scale of 60 were employed to extract vegetation information, as shown in Figure 8d.

### 3.2. Precision Verification

Accuracy evaluation is a crucial phase in the process of extracting information from remote sensing images. In this study, we used the fused WorldView-2 multispectral image as a reference, assuming the results of visual interpretation to be correct. The specific operation involved comparing specific pixels in the classified image with the known classified reference pixels. By using a confusion matrix, we calculated the overall accuracy, user accuracy, and producer accuracy of the classification. We randomly selected 400 points for assessing the accuracy of the remote sensing image feature extraction results. The evaluation of accuracy (Table 3) reveals that the Overall Accuracy (OA) is as high as 92.8%. This high precision indicates that our feature extraction produces reliable features to support our vulnerability index calculation.

**Table 3.** Precision evaluation.

| Category | Vehicle | Road | Building | Vegetation | Row Sum | User Accuracy |
|---|---|---|---|---|---|---|
| Vehicle | 85 | 3 | 12 | 0 | 100 | 85% |
| Road | 0 | 91 | 5 | 4 | 100 | 91% |
| Building | 0 | 4 | 96 | 0 | 100 | 96% |
| Vegetation | 1 | 3 | 0 | 94 | 100 | 94% |
| Column sum | 86 | 101 | 113 | 98 | 400 | 98% |
| Producer accuracy (%) | 99% | 90% | 85% | 96% | | |
| Overall accuracy (%) | | | | 92.8% | | |

The table above reveals that vehicle recognition has the lowest accuracy. This is primarily due to the distinct spectral characteristics of vehicles and the use of the object-oriented method for extracting dark and bright cars separately. Despite the high detection rate achieved through selecting appropriate domain values, the final accuracy rate was not as high due to the phenomenon of different objects sharing the same spectral characteristics in images. This results in the segmentation being sensitive to noise and other mixed objects. For instance, during the extraction process, dark objects such as road sign shadows, oil stains on the road, and vehicle shadows could be misidentified as vehicles. During the bright car extraction process, the highlighted reflective parts of partially shaped dark vehicles, parts of the divider, pavement marking lines, and others are incorrectly identified as closed vehicles. Additionally, some vehicles near the roadside and traffic lanes are merged with the divider or the edge line during edge detection. This results in open edges and non-detection, significantly impacting the detection rate. Furthermore, when the vehicle density on auxiliary road sections is high, with smaller spacing between vehicles and less exposed road surface, it becomes more difficult to distinguish foreground targets from the background, complicating vehicle target detection.

On the other hand, roads have more pronounced geometric and topological characteristics. They are displayed in clear lines, intersections, and networks, providing a more intuitive distribution of the road system. The method excels in extracting large-area transportation networks and hubs, offering a speed advantage, providing rapid, efficient support information for decision making in large-volume image analysis tasks. However, the extraction accuracy varies for different road targets. Roads without obstructions such as vegetation and building shadows are extracted with high precision. On the contrary, the extraction of occluded roads may result in misclassification or non-detection. Since

this road display accuracy is low and other ground objects cause significant interference, the extraction effect is poor, necessitating manual detail processing. Therefore, using these classified products as a reference, combined with the fieldwork team's daily reports, we have also conducted intensive post-processing to create as accurate as possible a feature inventory for the Tianshan District's urban area for the vulnerability index construction.

## 4. Discussions

### 4.1. Highly Vulnerable Micro-Geographic Units and Their Characteristics

Following Formulae (1)–(6), we calculated the entropy weight for each indicator as well as their contribution directions (positive or negative) to the urban vulnerability index and reported them in Table 4. We then calculated the *UVI* based on Formula (7) for all 30 micro-geographic units in the Tianshan District's built-up area and reported them in Table 5 and Figure 9.

**Table 4.** The weight of indicators ("+" suggests the index increases vulnerability, and "−" decreases).

| Micro-Geographic Unit Traffic Safety Vulnerability Index | Indicator Attribute | Weights |
|---|---|---|
| Vegetation patch density | + | 0.1240 |
| Vegetation extension index | − | 0.1247 |
| Vegetation Shannon index | − | 0.1252 |
| Average road vehicle density | + | 0.1253 |
| Vehicle density | + | 0.1253 |
| Road density | − | 0.1245 |
| Road complexity | + | 0.1266 |
| Building density | + | 0.1244 |

**Table 5.** Urban vulnerability index of Tianshan District, Urumqi, Xinjiang, China.

| Micro-Geographic Unit | Vegetation Patch Density | Vegetation Extension Index | Vegetation Shannon Index | Average Road Vehicle Density | Vehicle Density | Road Density | Road Complexity | Building Density | Vulnerability |
|---|---|---|---|---|---|---|---|---|---|
| X1 | 0.1243 | 0.1209 | 0.0209 | 0.1374 | 0.0840 | 0.1190 | 0.0874 | 0.1244 | 0.8183 |
| X2 | 0.0506 | 0.0984 | 0.0244 | 0.1211 | 0.0696 | 0.1189 | 0.0522 | 0.1031 | 0.6383 |
| X3 | 0.0126 | 0.0080 | 0.0766 | 0.0238 | 0.0544 | 0.1025 | 0.0353 | 0.0280 | 0.3412 |
| X4 | 0.0157 | 0.0004 | 0.1254 | 0.1029 | 0.0649 | 0.0533 | 0.0216 | 0.0055 | 0.3897 |
| X5 | 0.0962 | 0.1082 | 0.0836 | 0.1546 | 0.0946 | 0.0618 | 0.0189 | 0.1205 | 0.7386 |
| X6 | 0.1267 | 0.1098 | 0.0070 | 0.0913 | 0.0816 | 0.0834 | 0.0239 | 0.1279 | 0.6515 |
| X7 | 0.1220 | 0.1244 | 0.0139 | 0.1777 | 0.1098 | 0.0748 | 0.0880 | 0.0947 | 0.8053 |
| X8 | 0.1132 | 0.1052 | 0.0139 | 0.1069 | 0.1089 | 0.1058 | 0.0651 | 0.1028 | 0.7219 |
| X9 | 0.0952 | 0.0962 | 0.0348 | 0.0917 | 0.1253 | 0.0835 | 0.0750 | 0.0952 | 0.6969 |
| X10 | 0.0954 | 0.1184 | 0.0309 | 0.0574 | 0.0678 | 0.0402 | 0.1224 | 0.1066 | 0.6391 |
| X11 | 0.1002 | 0.1226 | 0.0958 | 0.0998 | 0.0691 | 0.0394 | 0.0630 | 0.1178 | 0.7077 |
| X12 | 0.0851 | 0.0455 | 0.0697 | 0.1030 | 0.1033 | 0.1165 | 0.0492 | 0.0660 | 0.6383 |
| X13 | 0.0867 | 0.0519 | 0.0453 | 0.0173 | 0.0169 | 0.0939 | 0.0049 | 0.0555 | 0.3723 |
| X14 | 0.1179 | 0.1146 | 0.0000 | 0.1143 | 0.0973 | 0.0724 | 0.1266 | 0.1244 | 0.7673 |
| X15 | 0.0721 | 0.0515 | 0.0906 | 0.1331 | 0.1298 | 0.1071 | 0.0628 | 0.0807 | 0.7277 |
| X16 | 0.1250 | 0.1062 | 0.0139 | 0.1253 | 0.0656 | 0.0152 | 0.0859 | 0.1011 | 0.6383 |
| X17 | 0.0671 | 0.0451 | 0.0766 | 0.0959 | 0.0987 | 0.0584 | 0.0649 | 0.0958 | 0.6025 |
| X18 | 0.0843 | 0.0911 | 0.0801 | 0.0383 | 0.0162 | 0.0729 | 0.0662 | 0.1090 | 0.5580 |
| X19 | 0.1150 | 0.1005 | 0.0557 | 0.0614 | 0.0862 | 0.0971 | 0.0082 | 0.0884 | 0.6127 |
| X20 | 0.0870 | 0.1004 | 0.0105 | 0.0497 | 0.0260 | 0.1192 | 0.0032 | 0.0809 | 0.4767 |
| X21 | 0.1096 | 0.0999 | 0.0418 | 0.0287 | 0.0256 | 0.0847 | 0.0114 | 0.0631 | 0.4648 |
| X22 | 0.0536 | 0.0515 | 0.1184 | 0.0080 | 0.0034 | 0.1179 | 0.0068 | 0.0738 | 0.4333 |
| X23 | 0.0697 | 0.1018 | 0.0244 | 0.0243 | 0.0199 | 0.0988 | 0.0040 | 0.0630 | 0.4059 |
| X24 | 0.0600 | 0.0679 | 0.0836 | 0.0063 | 0.0152 | 0.0931 | 0.0126 | 0.0453 | 0.3840 |
| X25 | 0.0790 | 0.1019 | 0.0418 | 0.0021 | 0.0173 | 0.1076 | 0.0042 | 0.0706 | 0.4245 |
| X26 | 0.0811 | 0.0761 | 0.0557 | 0.0067 | 0.0073 | 0.1157 | 0.0056 | 0.0728 | 0.4209 |
| X27 | 0.0682 | 0.0665 | 0.0906 | 0.0067 | 0.0114 | 0.1168 | 0.0016 | 0.1060 | 0.4677 |
| X28 | 0.0776 | 0.0600 | 0.0523 | 0.0571 | 0.0383 | 0.0871 | 0.0036 | 0.0481 | 0.4240 |
| X29 | 0.0641 | 0.0901 | 0.0697 | 0.0240 | 0.0104 | 0.1127 | 0.0068 | 0.0576 | 0.4355 |
| X30 | 0.0180 | 0.0011 | 0.0871 | 0.0000 | 0.0000 | 0.1245 | 0.0017 | 0.0003 | 0.2328 |

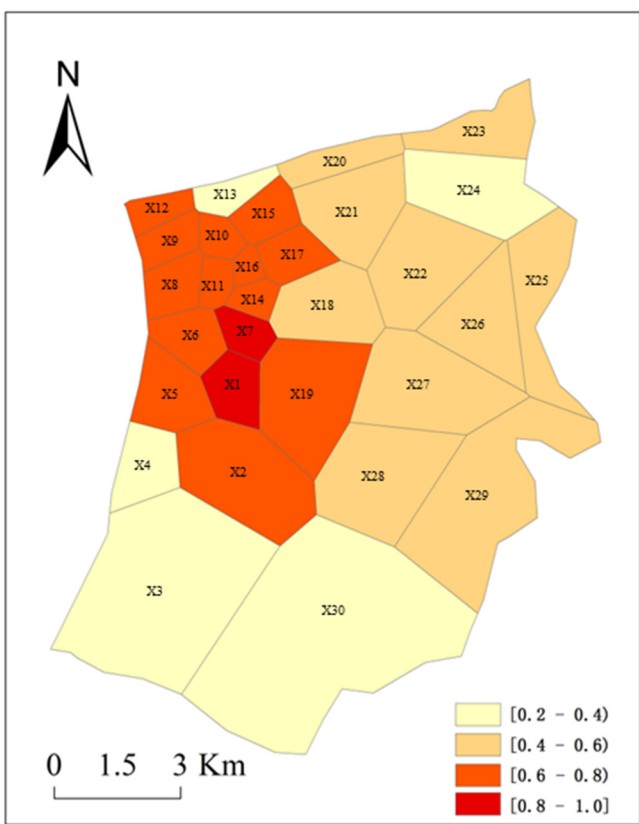

**Figure 9.** Urban public safety vulnerability.

From Figure 9 and Table 5, it becomes immediately apparent that two micro-geographic units, X1 (Erdaoqiao International Bazaar) and X7 (Golden Coin Mountain International Plaza), exhibit high vulnerability levels. These two units are geographically close (Figure 9). Their high vulnerability is primarily because of their high fragmentation in vegetation coverage and significant human activities (Table 5), as their road networks are complex and their building density is high, all of which contribute to their elevated vulnerability as we have defined in Section 3.

Moreover, the distribution of micro-geographic units with a higher vulnerability is not limited to X1 and X7. In fact, we identified a total of 13 micro-geographic units that fall into the somewhat high vulnerability category ($0.6 \leq UVI < 0.8$): X2, X5, X6, X8, X9, X10, X11, X12, X14, X15, X16, X17, and X19 (Table 5), and these micro-geographic units are spatially clustered to the northwestern part of the Tianshan District's built-up area (Figure 9). This cluster of micro-geographic units is characterized by heavy vehicular traffic, a dense road network, high building density, and intensive commercial activities (Figure 6), which are all significantly contributing to the urban vulnerability index as designed in our model. This cluster is also part of the central area of Urumqi city, neighboring Saybag District, where the People's Park and Hongshan Park are located. These two public parks often attract large amounts of traffic and people flow, especially in the summer times when the weather is amenable.

In addition, it was found that the road networks within these units are also quite complex. The high degree of road complexity index (number of intersections) signifies considerable traffic pressure due to the large number of turns and stops, resulting in poor road traffic capacity. Consequently, the road network's intricate nature and the subsequent transportation difficulties contribute to these units' high vulnerabilities.

This extensive list signifies a prevalent pattern of vulnerability that is closely related with heightened human activities (traffic and people), more complex road conditions (high vehicle density and excessive number of intersections), and highly fragmented vegetation

coverage. This pattern agrees well with the street level studies often observed in urban criminology studies [65–67], urban road accidents distribution [68], and general public safety studies [18,19,61]. To alleviate the vulnerability to common urban public safety threats like traffic accidents, environmental pollution, fire accidents, or even terrorist attacks, this demands comprehensive mitigation strategies and smart urban planning that would involve enhancing green connectivity, optimizing the road network for efficient transportation, and managing building density for better living conditions.

### 4.2. Medium- and Low-Level Vulnerable Micro-Geographic Units and Their Characteristics

The group of micro-geographic units that fall within the medium level of vulnerability category ($0.4 \leq UVI < 0.6$) is also clustered together, to the west of the core area of Tianshan District's built-up area, including units X18, X20, X21, X22, X23, X25, X26, X27, X28, and X29. The remaining five units, X3, X4, X13, X24, and X30, are among the lowest category of vulnerability in the Tianshan District ($0.2 \leq UVI < 0.4$). Not surprisingly, they are located in the outskirts of the Tianshan District, except for unit X13, which is located in the immediate south of Hongshan Park and is where many of the local college campuses are located, including the Corps University of Senior Citizen, East China Jiaotong University South Campus and Xinjinag Education College Sports Branch. This particular unit has relatively lower road complexity, vehicle density, and building density (Table 5), contributing to its low vulnerability in our defined framework.

These fifteen units of medium to somewhat-low vulnerability categories correspond to various landmarks and educational activity-centered neighborhoods of the Tianshan Districts: People's Square (the practical center of the District), Nanmen International City, Orthopedic Hospital, Changle Garden, Corps Hospital, Morning Light Garden, Xingfu Road Meteorological Community, Vocational University, Western Region International Trade City, Happiness Flower Court, Sun Moon and Stars Light Garden, Urumqi South Park, Urumqi Fourth Hospital, Xinjiang Education College, Water Park–Nanjiao Passenger Station, and South Campus of Xinjiang University.

Among these units, People's Square boasts of better greening, less vegetation fragmentation, lower density, and a relatively smaller building density. This is because as the district's de facto center, People's Square serves as a hub to connect to other parts of the district. The recent urban planning strategy to make the city more walkable and green starts from this center, allowing this micro-geographic unit to enjoy heightened greenery coverage with comparably lower building density and pedestrian friendly street layout. As a transportation hub, the road complexity is also comparably lower than neighboring micro-geographic units that connect to it.

Units like Nanmen International City, Orthopedic Hospital, Changle Garden, Corps Hospital, Morning Light Garden, Xingfu Road Meteorological District, Vocational University, Western Region International Trade City, Happiness Garden, Sun Moon Star Garden, Urumqi Fourth Hospital, and Xinjiang Education College exhibit fewer commercial activities, relatively low building density, broad roads, and good traffic conditions. These units, home to multiple high-rise residential quarters and educational facilities, demonstrate well-planned internal community greening, excellent vegetation coverage, comparably lower building density, and low vegetation fragmentation, contributing to their overall lower vulnerability in our defined framework.

Units such as Urumqi South Park, Water Park-Nanjiao Passenger Station, and the South Campus of Xinjiang University are distinctive for their very high vegetation coverage, low fragmentation, and high vegetation coverage. These areas contain fewer buildings because these are primarily recreational areas and are on the outskirts of the district. The road complexity is significantly lower, implying excellent road capacity and minimal vehicular traffic. Given their locations, these units experience less traffic pressure, and their built-up areas are often less densely distributed, which further mitigates their vulnerability.

*4.3. The Take-Home Message*

The current practice of utilizing remote sensing extracted features to understand micro-geographic units' vulnerabilities emphasizes that the interplay between commercial activity, building density, and road network intricacy is evident in their collective impact. High-intensity commercial activity coupled with a dense network of buildings and roads contributes to heightened vulnerability, suggesting the need for a more balanced approach in urban development that considers the interdependencies of these factors. On the contrary, higher vegetation coverage (greenery) with less fragmentation, broader roads with less complexity (less road intersections), and well-planned residential communities tend to be less vulnerable.

In addition, our analysis also suggests that in the Tianshan District of Urumqi, at least under our research framework of urban vulnerability, no micro-geographic units fall within the lowest vulnerability category. This result suggests that big cities like Urumqi are experiencing increasing vulnerability because of the intensified conflicts between rapid urbanization and limited land, infrastructure, and public service resources. To mitigate these vulnerabilities, it is crucial to implement sustainable urban planning strategies that balance urban growth with resource availability. Additionally, the development of resilient infrastructure and efficient public services can help alleviate the strain on existing resources. Regular vulnerability assessments, such as the one conducted in this study, can also play a pivotal role in identifying areas of concern and informing policy decisions.

These findings underscore the importance of effective urban planning in mitigating not only micro-geographic units' vulnerabilities but the overall urban vulnerabilities. Well-planned and greened areas fare better in terms of overall vulnerability, indicating the need for strategic urban planning. Areas with a high vegetation coverage and well-planned road networks further attest to the significance of green spaces and well-designed road infrastructure in reducing the pressure on urban spaces.

Overall, this analysis offers valuable insights for future urban planning and development strategies, emphasizing the need for careful consideration of green space allocation, road network design, and the density of commercial activities and buildings. These considerations will enable a more sustainable and resilient urban environment, where vulnerability is minimized, and the quality of life is enhanced.

## 5. Conclusions

Based on intensive fieldwork, remote sensing image analysis, and GIS operations, this study first establishes an important concept in urban vulnerability study: the micro-geographic units. The concept provides a very important viewpoint to study the highly granular cityscape in modern China that is often ignored primarily due to the difficulties of assembling data. The advent of remote sensing and big data analytics provide a great incentive to define and examine the cityscape with such granular details.

The study utilized multi-temporal satellite remote sensing data to interpret the spatial characteristics of cities, select different segmentation, and merge scales according to the sizes of various ground objects and established rule sets based on the spectra, shapes, and ductility of different ground objects. As a result, we were able to create a comprehensive index set to assess the vulnerability of micro-geographic units.

We conducted a vulnerability measurement experiment in the built-up area of the Tianshan District, Urumqi, Xinjiang. The vulnerability scores of the micro-geographic units were calculated and mapped, providing an in-depth understanding of the specific aspects of urban vulnerability within the district. We found that the urban centers that are characterized by heavy vehicular traffic, a dense road network, high building density, and intensive commercial activities tend to be of higher vulnerability. Both our fieldwork and the analytical results suggest that the high degree of road complexity (number of intersections) signifies considerable traffic pressure and increased vehicular–pedestrian interaction risks due to the large number of turns and stops. Consistent greenery is often a sign of lowered vulnerability, but fragmented greenery (higher patch index) might actually

increase the micro-geographic units' vulnerabilities because the fragmented greenery often suggests a high density of built-up areas or other types of impervious surfaces.

Our findings underscore the potential of integrating fieldwork, remote sensing image analysis, and GIS operations in urban vulnerability research. The micro-geographic unit approach, fostered by advanced technology, offers a new level of granularity and detail in assessing urban vulnerability. It sets the stage for more efficient and strategic urban planning initiatives and can inform decision-making processes regarding sustainable urban development. Furthermore, our results can serve as a blueprint for future studies focusing on other regions, thus broadening the scope and applicability of this approach. As we continue to refine this methodology, we believe it will offer crucial insights into mitigating urban vulnerabilities and fostering resilient urban environments.

Still, the current work is not without its limitations. As we have argued in the beginning of defining the micro-geographic unit, it is a highly dynamic and local-knowledge-dependent concept for efficient urban public safety strategic planning. This suggests that adaptation of the concept to other areas of study requires similarly intensive fieldwork and local knowledge, which might not be readily available. In addition, integrating more remote sensing image information to build the micro-geographic unit's index set will be of higher priority. With advanced remote sensing and social sensing information extraction techniques, we aim to integrate a more comprehensive index set for our micro-geographic units in the future and attempt to apply the concept and data collection strategies to other cities as well.

**Author Contributions:** Conceptualization, J.Z. and D.Y.; Data curation,; Formal analysis, J.Z. and D.Y.; Funding acquisition, D.Y.; Investigation, J.Z., D.Y., C.H. and Z.W.; Methodology, J.Z. and D.Y.; Project administration, J.Z. and D.Y.; Resources, J.Z. and D.Y.; Software, J.Z. and C.H.; Supervision, J.Z. and D.Y.; Validation, J.Z. and D.Y.; Visualization, J.Z. and D.Y.; Writing—original draft, J.Z. and D.Y.; Writing—review and editing, D.Y. All authors have read and agreed to the published version of the manuscript.

**Funding:** This research was funded by the Chinese National Nature Science Foundation Regional Grant "Studies of urban vulnerability based on micro-geographic units via spatial data analysis and geocomputation" (grant number 41461035).

**Data Availability Statement:** Not applicable.

**Conflicts of Interest:** The authors declare no conflict of interest.

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
