# Peer review of "Urban Vulnerability Analysis Based on Micro-Geographic Unit with Multi-Source Data—Case Study in Urumqi, Xinjiang, China"

_remotesensing, doi:10.3390/rs15163944_

Round 1
Reviewer 1 Report
The authors took Urumqi Tianshan District as the research area, combined with the field survey data, considers the urban public safety vulnerability, constructed micro-geographical units, established a reasonable evaluation index system of urban vulnerability, and used the entropy method to take the micro-geographical unit as the Research Unit to study the quantification of urban public safety. The research is interesting and helpful for future research in the field. However, there are still several weaknesses in the present manuscript, which I will now elaborate on. Below, I added some comments. More detailed suggestions can be found in the attached file.
Point 1: I suggest the authors should add the quantitative research results to Abstract. In addition, some content in the abstract should be case-sensitive, such as “urban public Safety vulnerability”, and “Research Unit”, should be “urban public safety vulnerability”, and “research unit”.
Point 2: Add " Urban public safety" in Keywords.
Point 3: The necessity and innovation of the article should be obviously presented to the introduction. The authors should improve them in the later version.
Point 4: Please state your research question clearly at the end of the introduction section. This will help the readers to grasp your key scientific question.
Point 5: In Section 3.1, “Yuan” should be “CNY”.
Point 6: Table 1 should include indicator attributes (positive indicators or negative indicators).
Point 7: In Section 4.4, “low level (0.2 < CPSV ≤ 0)” should be “low level (0.0 < CPSV ≤ 0.2)”.
Point 8: The Authors did not verify obtained results with other authors and the actual state of knowledge. The Authors should provide some information of advantages and disadvantages of performed analysis.
Point 9: Add the uncertainties or limitations of this study and how further research will be continued in the discussion.
Point 10: In case of conclusion, there should find the main achievements that could be obtained from the conducted analysis. And it would be better to provide more concise conclusions in the end.

Moderate editing of English language required
Author Response
Reviewer 1
The authors took Urumqi Tianshan District as the research area, combined with the field survey data, considers the urban public safety vulnerability, constructed micro-geographical units, established a reasonable evaluation index system of urban vulnerability, and used the entropy method to take the micro-geographical unit as the Research Unit to study the quantification of urban public safety. The research is interesting and helpful for future research in the field. However, there are still several weaknesses in the present manuscript, which I will now elaborate on. Below, I added some comments. More detailed suggestions can be found in the attached file.
Response: Thank you for your critical comments and suggestions. We strive to address your concerns adequately and hope the revision will gain your approval and support.
Point 1: I suggest the authors should add the quantitative research results to Abstract. In addition, some content in the abstract should be case-sensitive, such as “urban public Safety vulnerability”, and “Research Unit”, should be “urban public safety vulnerability”, and “research unit”.
Response: Thank you for your meticulous comment. We went through the entire manuscript and checked all discrepancies to ensure that any such problems are checked and corrected.
Point 2: Add " Urban public safety" in Keywords.
Response: Added.
Point 3: The necessity and innovation of the article should be obviously presented to the introduction. The authors should improve them in the later version.
Response: We have modified the introduction section, with additional literature review, to provide a clearer presentation of the necessity and innovation of the current research. We hope the revision will gain your support and approval.
Point 4: Please state your research question clearly at the end of the introduction section. This will help the readers to grasp your key scientific question.
Response: Before stating the structure of the manuscript, we have made revisions to make the research questions and objectives of the research clear now.
Point 5: In Section 3.1, “Yuan” should be “CNY”.
Response: Corrected.
Point 6: Table 1 should include indicator attributes (positive indicators or negative indicators).
Response: Added
Point 7: In Section 4.4, “low level (0.2 < CPSV ≤ 0)” should be “low level (0.0 < CPSV ≤ 0.2)”.
Response: We changed the CPSV to UVI and corrected the numbering issue.
Point 8: The Authors did not verify obtained results with other authors and the actual state of knowledge. The Authors should provide some information of advantages and disadvantages of performed analysis.
Response: Thank you for this critical comment. We have done some comparisons with other studies when we develop the concept of microgeographic units in sections 2.3. We compared the results from our study based on the microgeographic units with urban criminology studies, road traffic accident studies and general urban public safety studies, and found our results agree well with studies in those fields. We have added the statement in section 4.1. We hope the revision will gain your approval and support.
Point 9: Add the uncertainties or limitations of this study and how further research will be continued in the discussion.
Response: We have added uncertainties and limitations at the end of the conclusion section to address potential limitations and future directions. We hope this revision will gain your support.
Point 10: In case of conclusion, there should find the main achievements that could be obtained from the conducted analysis. And it would be better to provide more concise conclusions in the end.
Response: Thank you for your suggestion. We have now added a concise summary of what we found through the current study and added that to the conclusion section. We hope this revision will gain your support.
Detailed comments:
Abstract: Some necessary background information should be included here.
Response: We added the inspiration from our current study (streetscape analysis in urban criminology studies) in the abstract. We hope this would address your concerns.
Key words: remove this semicolon
Response: Another key word “Urban public safety” is added after the semicolon per your suggestion.
Page 1: In fact, I can hardly get your point in P1. Additionally, some necessary citations should be added here. Logical Relationships between these two paragraphs should be enhanced.
Response: Per your suggestion, we have added relevant studies here and improved the logical relationships between the paragraphs.
Page 2: Avoid etc. Actually, it is not a scientific expression. similarly hereinafter
Response: All removed. Thank you.
Please provide a bit more big-picture motivation of how your analyses benefit society and how they have evolved over the past decade. However, from my point of view, the article does not provide a sufficiently thorough review of the issue under study. There are good references for the study techniques, but the paper is missing a "big-picture" introduction with some references in my opinion. I suggest that the authors should do a better analysis of the literature. It seems that the bulk of the text is a sort of compilation of statements in the individual articles cited. It would be better, I think, to extract ideas from individual articles and tie them together into a more fluid and conceptually homogeneous text. As it is, the text looks rather clumsy.
In my opinion, they should be merged and reduced to highlight your point.
Here, the authors should emphasize their objectives rather than framework.
Research gaps, objectives of the proposed work should be clearly justified before the problem formulation section. This paper includes some little useful information and the main objectives of the study is not well defined. Problem statement is not clear and the objectives are obscure. Furthermore, the paper lacks a very clear and good justification for what is new and innovative about this case or this case or this approach.
Response: We have restructured the last few paragraphs with additional literature research to provide a smoother read. We hope this revision will gain your support.
Page 3: In fact, this section is crucial, because it illustrate your basic study object. However, this section lacks the necessary structure.
Until now, I can hardly get your expression.
Response: Thank you for the crucial comments. We have re-read few times of this section and agree with your observations. We have added subtitles in the section and separated the lump sum of the paragraphs to more structured, readable paragraphs. We hope this revision will make this crucial section clearer and well-structured.
Page 5: The map should be reorganized. In addition, its quality should be further improved.
Response: Maps are remade to make them clearer.
Maybe you refer to framework?
Response: Yes, we have added framework in the sentence as well
Page 6: Where is Fig 1.a
Response: Figure 1.a is separated from the other two figures. After further examination, we feel that Figure 1.a is not entirely necessary. We have removed Figure 1a to make the illustration clearer.
What is EWM
Response: We have added an illustration in the caption of figure 2 that EWM stands for entropy weighting method, which we detailed in the method section.
Page 21: Conclusion section should be more specific and concise.
Response: Thank you for the suggestion. We have significantly revised the conclusion section to make it more specific and concise. We hope the revision will gain your support and approval.
Reviewer 2 Report
Thanks to the authors for their articles. Studying urban areas' vulnerability is an important and current issue. This paper has a certain degree of innovation in the case of micro-geographic units, and the research is also very interesting. There are the following issues, and it is recommended to modify them.
-The importance of the study should be emphasized by making more references to the international literature. In addition, it is necessary to write a separate discussion section, which will show the richness of the study.
-In the discussions, I would like a detailed explanation of the other possible factors (except revealed factors such as a lack of green spaces, poor urban planning, dense building development, and traffic issues) on public security vulnerability.
-There is no direct statistical number of events/crimes or cases documented in the micro-geographic units of Tianshan District, Urumqi City against public safety.
If these improvements are made, the study has the potential to be published in the journal.
Author Response
Thanks to the authors for their articles. Studying urban areas' vulnerability is an important and current issue. This paper has a certain degree of innovation in the case of micro-geographic units, and the research is also very interesting. There are the following issues, and it is recommended to modify them.
Response: Thank you for your constructive suggestions. We will hope to adequately address your concerns and hope you gain your support and approval.
-The importance of the study should be emphasized by making more references to the international literature. In addition, it is necessary to write a separate discussion section, which will show the richness of the study.
Response: Thank you for the suggestion. We have revised the introduction section and the result section and added more references to connect more with the international research community. We then further separated the results and discussion into two separate sections. We hope the modifications will gain your approval and support.
-In the discussions, I would like a detailed explanation of the other possible factors (except revealed factors such as a lack of green spaces, poor urban planning, dense building development, and traffic issues) on public security vulnerability.
Response: Thank you for the suggestion. We have highlighted how these factors are influencing the urban vulnerability index in our current modeling framework. We have also compared the work in the current study with other similar studies to draw attention to how these factors might influence the urban vulnerability at the microgeographic unit level. We hope the revision will gain your support and approval.
-There is no direct statistical number of events/crimes or cases documented in the micro-geographic units of Tianshan District, Urumqi City against public safety.
Response: Yes, you are correct that direct numbers of events/crimes or cases are not documented in the micro-geographic units. This is because the micro-geographic units are dynamically defined and modified based on local knowledge instead of any officially established boundaries. In addition, we have relied solely on remote sensing and field survey to establish our microgeographic unit data set, which unfortunately, will not allow us to collect relevant statistical numbers of events/crimes or cases. We hope this explanation will gain your support and understanding.
If these improvements are made, the study has the potential to be published in the journal.
Response: Thank you again for your thorough review.
Reviewer 3 Report
Presented manuscript concentrated on the urban vulnerability analysis based on micro-geographic units in Urumqi, Xinjiang (China).
For this analyses, the Author/Authors used the multi-source data involving high-resolution remote sensing. The Authors used WorldView-2 remote sensing imagery and the object-oriented classification method to extract and evaluate features related to vehicles, roads, buildings, and vegetation for each unit. This study introduces a novel approach to urban public safety analysis. Authors have developed urban vulnerability evaluation index.
The topic is interesting and actual. The main ideas of the article and the questions posed and analyzed by the Author/Authors are relevant for the Remote Sensing.
The proposed research methodology, data and tools used can be used by different economic and planning sectors at different levels, in all areas where knowledge of urban public safety is valuable.
After reading the article, the following comments came to mind:
(1) The Introduction section (line 89-99) should be strengthened to clearly summarise what has been found and what has not, in order to demonstrate the contribution of this study to the development of this topic.
(2) Chapter 2 (line 135-200) contains methodological assumptions, which should be included in the chapter of method. Some of them should be moved to chapter 3.2. Lines 144 -The Authors wrote about the field research, but did not write what specifically the field research consisted of.... In line 148, the authors state that for the first time candidate locations for the study were selected during the pre-study conference, and in lines 150-151 that all members contributed comments based on their own observations and experiences .... But in this part of the manuscript we learn nothing on what criteria this field survey was conducted.... general principles are given (line 152). Later (line 166-167) the authors write about adjusting the survey strategy and eventually reducing the number of locations.
The article is not well constructed (Chapter 2, 3 and 4). I propose to restructure the manuscript because it is not clear. I suggest introducing essential chapters such as for example 1. Introduction, 2. Materials and method, 2.1. Study Area, 2.2. Datasets and Sources, 2.3. Methods, 3. Results and Discussions, 4. Conclusion.
(3) The chapter on methodology is not easy to read. It is not prepared in a synthetic manner... for example, Chapter 3.3 (line 295) the Authors write „For this purpose, we attempt to develop a so-called urban public safety vulnerability index based on specific micro-geographic unit elements” and only in Chapter 4.3 Authors discussed the construction of vulnerability model….
(4) Lines 200-201 should not be separated by a figure.
(5) Line 246-248; It is not clear what these corrections consisted of.... Did the Authors mean that after the micro-geographical units were delineated using the Voronoi method, these boundaries were then corrected with respect to natural boundaries such as transportation routes, rivers? The role of surveyors is not clear here....
(6) In line 252-253; The Authors wrote „Defining these micro-geographic units clearly forms the basis of our research and represents a focal point and innovative aspect of our work”; But in very general terms, in lines 249-251, they write about factors such as demographic, economic or environmental, without indicating the determining indicators in these particular categories.
(7) Please mark the points with their location and their designation (number) on figure 3 (b), it will be easier to read.
(8) Data Sources section 3.4.; Additionally I suggest making a table listing all the datasets used in the analyses, their sources and characteristics, their scale of compilation, their validity date …What about social-economic data? Was it descriptive data referenced to administrative boundaries or in grid form ?
(9) Chapter 4.3.2, line (444-445); Have the Authors checked that the indicators do not correlate with each other and do not carry the same information...?
(10) In Chapter 4.3.3.; It would be useful to have a summary of these factors in the form of a table, their characteristics, definition, unit, type whether positive or negative, and calculated weight. This would be more readable….
(11) Line 468-471 (??? into five levels: 0.8 ≤ ??? < 1.0); In my opinion the authors should have, after assigning weights and calculating the UVI, standardization the UVI again so that its spread is from [0-1].
(12) No reference of the formulas used to the literature.
(13) Line 470-471; Why did the Authors use class division with equal ranges? The division into classes should be based on the statistical properties of the UVI variable, i.e. mean and standard deviation. Adopting a division into equal class ranges may result in some classes being empty!
(14) Figure 8; It is a pity that the Authors did not show the vulnerability indicators of the micro-geographical unit on the maps (from Table 1). The feature extraction results do not tell us about the density of the phenomena analysed.
(15) In figure 9, we see four classes and not five , the reasons for this situation are described in my comments (11) and (13).
(16) Did the Authors analyse the characteristics of these individual class types, which of these indicators were decisive for the membership of individual micro-geographical units in a particular class?
(17) Chapter 5.2 Precision Verification; Some of these methodological assumptions should be in the methodology section.
(18) I suggest reviewing the Discussion; Chapter 5 is not a scientific discussion. This chapter includes results and is a summary and assumptions for future research. There is a lack of reference of the results obtained to other researchers, which is important in scientific publication ! Already in the Introduction, the Authors announce an comprehensive discussion (line 97-98 „ ….. with a comprehensive discussion”).
(19) With reference to the text read, further questions arises... For other regions of China and the world, would the proposed indicators to be modified due to the specificities of the region, the country...? In which areas, geographical regions do the Authors intend to conduct empirical research, or only in the Tianshan District?
Synthetic indicator UVI [0.0-1.0] for each micro-geographic units (in the form of a latent variable) was constructed by the summation of standardized values [0-1] of partial indicators and the maximum and minimum stimulant and destimulant values that can occur when analysing this phenomenon will not always be of the same magnitude. So a UVI of, say, 0.8-1.0 in one place will not always be the same UVI (0.8-1.0) in another place….., only from among the achievable values of both the maximum and minimum in a given area.
Author Response
Presented manuscript concentrated on the urban vulnerability analysis based on micro-geographic units in Urumqi, Xinjiang (China).
For this analyses, the Author/Authors used the multi-source data involving high-resolution remote sensing. The Authors used WorldView-2 remote sensing imagery and the object-oriented classification method to extract and evaluate features related to vehicles, roads, buildings, and vegetation for each unit. This study introduces a novel approach to urban public safety analysis. Authors have developed urban vulnerability evaluation index.
The topic is interesting and actual. The main ideas of the article and the questions posed and analyzed by the Author/Authors are relevant for the Remote Sensing.
The proposed research methodology, data and tools used can be used by different economic and planning sectors at different levels, in all areas where knowledge of urban public safety is valuable.
Response: Thank you for your meticulous review. We will strive to address your concerns adequately to gain your further support and approval.
After reading the article, the following comments came to mind:
(1) The Introduction section (line 89-99) should be strengthened to clearly summarise what has been found and what has not, in order to demonstrate the contribution of this study to the development of this topic.
Response: Thank you for your constructive suggestion. We have modified the introduction section to provide a clearer statement of the contribution the current study has made to the remote sensing and urban public safety community.
(2) Chapter 2 (line 135-200) contains methodological assumptions, which should be included in the chapter of method. Some of them should be moved to chapter 3.2. Lines 144 -The Authors wrote about the field research, but did not write what specifically the field research consisted of.... In line 148, the authors state that for the first time candidate locations for the study were selected during the pre-study conference, and in lines 150-151 that all members contributed comments based on their own observations and experiences .... But in this part of the manuscript we learn nothing on what criteria this field survey was conducted.... general principles are given (line 152). Later (line 166-167) the authors write about adjusting the survey strategy and eventually reducing the number of locations.
The article is not well constructed (Chapter 2, 3 and 4). I propose to restructure the manuscript because it is not clear. I suggest introducing essential chapters such as for example 1. Introduction, 2. Materials and method, 2.1. Study Area, 2.2. Datasets and Sources, 2.3. Methods, 3. Results and Discussions, 4. Conclusion.
Response: Thank you for the suggestion. We have discussed it among the authors, and we agree a better structure can be achieved. We then merged the methods test together, and modify the entire structure based on your suggestion. We hope the revision will gain your approval and support.
(3) The chapter on methodology is not easy to read. It is not prepared in a synthetic manner... for example, Chapter 3.3 (line 295) the Authors write „For this purpose, we attempt to develop a so-called urban public safety vulnerability index based on specific micro-geographic unit elements” and only in Chapter 4.3 Authors discussed the construction of vulnerability model….
Response: After the restructuring of the manuscript, now the flow of the methodology part shall be better structured than the original manuscript. Thank you for your constructive suggestions the make the manuscript better.
(4) Lines 200-201 should not be separated by a figure.
Response: This is due to the Word’s handling of figures. We have put all the elements of Figure 1 together and hopefully it will prevent such mishaps in the revision.
(5) Line 246-248; It is not clear what these corrections consisted of.... Did the Authors mean that after the micro-geographical units were delineated using the Voronoi method, these boundaries were then corrected with respect to natural boundaries such as transportation routes, rivers? The role of surveyors is not clear here....
Response: We have added clarification here: “Specifically, we adjust the Voronoi boundaries using the nearest streets so that the identified microgeographic units are consistent with street separated communities.” We hope this clarification addresses your concern.
(6) In line 252-253; The Authors wrote „Defining these micro-geographic units clearly forms the basis of our research and represents a focal point and innovative aspect of our work”; But in very general terms, in lines 249-251, they write about factors such as demographic, economic or environmental, without indicating the determining indicators in these particular categories.
Response: Thank you for pointing this out. We realize that the microgeographic units defined in our study have a weak capacity to identify homogeneity among demographic, economic or environmental factors. Instead, the primary purpose of the microgeographic units is to establish a streetscape view of urban public safety measures that can be derived from remote sensing technologies. We have removed the general terms. We hope the revision will gain your approval and support.
(7) Please mark the points with their location and their designation (number) on figure 3 (b), it will be easier to read.
Response: Thank you for your suggestion. The survey locations are provided in Figure 1(c). Figure 3(b) is provided with an illustration of the Voronoi algorithm and how it is applied to the Tianshan District. Putting the locations and designations on Figure 3(b) makes the map quite a bit crowded. We hope this explanation addresses your concerns.
(8) Data Sources section 3.4.; Additionally I suggest making a table listing all the datasets used in the analyses, their sources and characteristics, their scale of compilation, their validity date …What about social-economic data? Was it descriptive data referenced to administrative boundaries or in grid form ?
Response: Thank you for your suggestion. The purpose of this study is to rely on remote sensing feature extraction to build an element set for a non-traditional “microgeographic units.” The data is solely built from remote sensing image. Traditional socioeconomic information is not included in the index set because the boundaries of the microgeographic units do not agree with either the administrative boundaries or regular grid demarcation. We added a clarification there to clearly indicate this point. We hope the clarification addresses your concerns adequately.
(9) Chapter 4.3.2, line (444-445); Have the Authors checked that the indicators do not correlate with each other and do not carry the same information...?
Response: Thank you for your meticulous review. We did a correlation analysis and indeed found that the vegetation indices and vehicle density indices are correlated. However, the entropy weighing method is able to effectively overcome the influence of such correlation. We have added clarification to bring this point up front. We hope the revision adequately addresses your concerns.
(10) In Chapter 4.3.3.; It would be useful to have a summary of these factors in the form of a table, their characteristics, definition, unit, type whether positive or negative, and calculated weight. This would be more readable….
Response: We added relevant information in Table 1, and hopefully this will address your concern.
(11) Line 468-471 (??? into five levels: 0.8 ≤ ??? < 1.0); In my opinion the authors should have, after assigning weights and calculating the UVI, standardization the UVI again so that its spread is from [0-1].
Response: Per our calculations, the UVI falls within the range of [0 - 1]. This is guaranteed because the weights are summed to 1, and the standardized Sijs are in the range of [0 – 1]. Hence the sum of the multiplication of these two terms is within the range of [0 – 1]. We hope this explanation addresses your concerns.
(12) No reference of the formulas used to the literature.
Response: Thank you for your meticulous review. For some reason, the reference was deleted during the editing process. We have added them back.
(13) Line 470-471; Why did the Authors use class division with equal ranges? The division into classes should be based on the statistical properties of the UVI variable, i.e. mean and standard deviation. Adopting a division into equal class ranges may result in some classes being empty!
Response: The decision was made based on literature research, but more importantly, based on the discussion among the research team, especially the local students who have participated in the fieldwork and also lived their lives in the study area. An equal interval seems to summarize the feelings of local knowledge base well. Without further complicating the analysis, and under the principle of following “local-knowledge”, we proceeded with the equal interval classification. We have presented the reason in the text and hope this clarifies our purpose.
(14) Figure 8; It is a pity that the Authors did not show the vulnerability indicators of the micro-geographical unit on the maps (from Table 1). The feature extraction results do not tell us about the density of the phenomena analysed.
Response: Figure 8 merely shows the feature extraction results of the entire Tianshan District. The vulnerability indicators are derived from these results. We reported the final vulnerability index in Figure 9, and the individual index’s values in Table 5. We hope this information is adequate to demonstrate the purpose of the current study without adding too much redundant information
(15) In figure 9, we see four classes and not five , the reasons for this situation are described in my comments (11) and (13).
Response: Thank you for your meticulous review. We are indeed aware that there are only four classes (namely, the lowest level class is not present). We leave this on purpose to showcase that at least in our framework of measuring urban vulnerability based on microgeographic units, there are no such units that have the lowest vulnerability. We added an articulation in the discussion section to suggest just that and provides our view for balanced urban planning. We hope the revision will adequately address your concerns.
(16) Did the Authors analyse the characteristics of these individual class types, which of these indicators were decisive for the membership of individual micro-geographical units in a particular class?
Response: Thank you for your suggestions. Yes, the entropy weighting method addresses the concern here. Based on the entropy weighting method, it seems all seven indicators play similar roles in determining the class types, after considering the overlapping information among the indicators.
(17) Chapter 5.2 Precision Verification; Some of these methodological assumptions should be in the methodology section.
Response: We have restructured the entire manuscript based on your previous suggestions. Now it is in the result section.
(18) I suggest reviewing the Discussion; Chapter 5 is not a scientific discussion. This chapter includes results and is a summary and assumptions for future research. There is a lack of reference of the results obtained to other researchers, which is important in scientific publication ! Already in the Introduction, the Authors announce an comprehensive discussion (line 97-98 „ ….. with a comprehensive discussion”).
Response: Result and discussion sections are separated and re-written. We hope the revision addresses your concerns.
(19) With reference to the text read, further questions arises... For other regions of China and the world, would the proposed indicators to be modified due to the specificities of the region, the country...? In which areas, geographical regions do the Authors intend to conduct empirical research, or only in the Tianshan District?
Response: Thank you for this keen observation. We have added a comment in the conclusion section that because the definition of microgeographic units is a highly dynamic and local knowledge-dependent concept, adaptation of the concept to other areas of study requires similarly intensive fieldwork and local knowledge, which might not be readily available, though possible.
Synthetic indicator UVI [0.0-1.0] for each micro-geographic units (in the form of a latent variable) was constructed by the summation of standardized values [0-1] of partial indicators and the maximum and minimum stimulant and destimulant values that can occur when analysing this phenomenon will not always be of the same magnitude. So a UVI of, say, 0.8-1.0 in one place will not always be the same UVI (0.8-1.0) in another place….., only from among the achievable values of both the maximum and minimum in a given area.
Response: Under the current design of the index set, we believe that the standardization of the indices and entropy weighting method are effective to solve this problem because the maximum (minimum) achievable values are assumed to be the same for all the microgeographic units. We hope this clarifies the calculation and gain your approval.
Thank you very much for your meticulous review, which helps us to improve the flow and organization of the manuscript.